# Co-Designing an Inclusive Stakeholder Engagement Strategy for Rehabilitation Technology Training Using the I-STEM Model

**DOI:** 10.3390/ijerph23010013

**Published:** 2025-12-20

**Authors:** Holly Blake, Victoria Abbott-Fleming, Asem Abdalrahim, Matthew Horrocks

**Affiliations:** 1School of Health Sciences, University of Nottingham, Nottingham NG7 2HA, UK; 2NIHR Nottingham Biomedical Research Centre, Nottingham University Hospitals NHS Trust, Nottingham NG7 2UH, UK; 3The British Pain Society, London WC1R 4SG, UK; 4Princess Salma Faculty of Nursing, Al al-Bayt University, Mafraq 25113, Jordan

**Keywords:** rehabilitation technologies, inclusion, accessibility, sensory impairment, cognitive impairment, physical disability, universal design, assistive technology

## Abstract

**Highlights:**

**Public health relevance—How does this work relate to a public health issue?**
Equitable access to rehabilitation technologies is essential for reducing health inequalities, particularly for people with sensory, cognitive, or physical impairments.Training that is inaccessible or poorly designed can limit the effectiveness of rehabilitation technologies, undermining recovery outcomes and long-term independence.

**Public health significance—Why is this work of significance to public health?**
This study identifies systemic barriers that prevent diverse populations from benefiting fully from rehabilitation technologies.It provides evidence-based priorities for designing inclusive training approaches that support equitable participation in rehabilitation.

**Public health implications—What are the key implications or messages for practitioners, policy makers and/or researchers in public health?**
Practitioners should adopt training models that address physical, sensory, and cognitive accessibility to ensure all users can engage effectively with rehabilitation technologies.Policymakers and researchers can use the co-created stakeholder engagement strategy to guide future intervention development and strengthen inclusive rehabilitation practices.

**Abstract:**

**Background:** Rehabilitation technologies, including assistive devices, adaptive software, and robotic systems, are increasingly integral to contemporary rehabilitation practice. Yet, ensuring that training in their use is inclusive and accessible remains a critical challenge. **Methods:** This study reports findings from patient and public involvement (PPI) activities conducted by the National Institute for Health and Care Research (NIHR) HealthTech Research Centre in Rehabilitation. Fifteen contributors participated, comprising rehabilitation professionals and educators, individuals with lived experience of serious illness, injury, or disability requiring rehabilitation, and technology innovators. The purpose of these activities was to identify the factors necessary to ensure that training in rehabilitation technologies is equitable for people with sensory, cognitive, and physical impairments. **Findings:** Contributors highlighted a series of priority domains that together capture the breadth of challenges and opportunities in this area. These included the need to address physical, sensory, and cognitive accessibility; to foster participation, motivation, and engagement; to strengthen instructional design and delivery; to ensure technological accessibility and integration; to enhance staff training and competence; and to embed participant-centred and policy approaches. Contributions in these domains were synthesised into thematic categories that provide a structured understanding of the training requirements of rehabilitation technology recipients. **Evaluation:** The PPI process was evaluated using the Guidance for Reporting Involvement of Patients and the Public (GRIPP2) Short Form, supplemented by an evaluation survey. This dual approach ensured that the contributions were systematically documented and critically appraised. **Implications:** Guided by implementation science, the principal output of this work was a co-created stakeholder engagement strategy, structured using the Implementation STakeholder Engagement Model (I-STEM). This plan will serve as a foundation for future research exploring the education and training needs of diverse stakeholder groups, thereby contributing to the development of more inclusive and effective rehabilitation technology training practices.

## 1. Introduction

The United Kingdom (UK) is increasingly embracing rehabilitation technologies, from mobility aids to smart home systems, to boost independence and ease the burden of healthcare demands [1]. Yet adoption remains uneven due to challenges like limited stakeholder training, lack of professional confidence, and low integration into practice [2]. Collaboration among professionals, educators, developers, and those with lived experience is key to sharing insights and identifying training needs [3]. Patient and public involvement (PPI) is the essential first step [4].

Rehabilitation technologies, ranging from mobility aids and robotic exoskeletons to digital applications and virtual reality platforms, play a transformative role in supporting recovery and independence for individuals with disabilities [1,5,6]. Yet, for these technologies to be effective, users must receive adequate and inclusive training [2]. Individuals with sensory, cognitive, or physical impairments often encounter barriers that limit their ability to access or benefit from such instruction [7].

This paper presents PPI contributions, which outline the essential factors that must be considered to ensure that training in the use of rehabilitation technologies is inclusive, accessible, and responsive to diverse needs. This PPI initiative was conducted in collaboration withthe NIHR HealthTech Research Centre in Rehabilitation and was undertaken to investigate stakeholder perspectives on training needs surrounding rehabilitation technologies. Specifically, the work sought to elicit views on two domains: first, the inclusivity and accessibility of training provision for individuals with sensory, cognitive, and physical impairments; second, the specific patient education and training needs associated with acquiring skills in the use of rehabilitation technologies. The principal outcome of this activity was the co-development of a stakeholder engagement strategy, which will serve as a framework for guiding future collaboration with public contributors and for informing the identification of education and training requirements in the rehabilitation technology field.

## 2. Materials and Methods

### 2.1. Definitions

‘Rehabilitation professionals’ were defined as ‘health or care workers who help individuals restore physical, mental, or cognitive function and improve quality of life following injury, illness, or disability’. ‘Health educators’ worked in higher education settings, were registered healthcare professionals and had responsibility for education delivery on rehabilitation topics. ‘Innovators’ were technology inventors/creators. ‘Patients’ were people with lived experience of serious injury, illness or disability requiring rehabilitation. Our PPI-partner was considered an equal member of the research team, supported by the project lead, and is a co-author on this paper. Reference to the ‘project team’ therefore includes our PPI-partner. Those who engaged in PPI were not research participants so are described as ‘PPI-contributors’. We adopted the NIHR definition of co-production in research as “an approach in which researchers, practitioners and the public work together, sharing agency and responsibility from the start to the end of the project, including the generation of knowledge” [8]. Per contributor preference, in our consultations we used ‘person-first’ language, which emphasises the individual before the disability, framing disability as one aspect of a person’s identity, not the defining feature.

### 2.2. PPI and Co-Production Process

The PPI process was informed by the UK Standards for Public Involvement [9], which advocate that public involvement partnerships are accessible and include a range of people and groups. Accessibility was prioritised by following UK government guidance on inclusive communication [10], including flexible engagement options, and building trust with PPI-contributors through co-production.

In the first stage, contributors were approached through professional networks and support groups. They were invited to share their views via asynchronous (email, text, survey, and video) and synchronous (meetings and calls) formats, per contributor preference. Support was available for accessibility needs and scheduling, including evenings and weekends. To reach under-represented groups (e.g., in terms of ethnicity and lower socioeconomic status), we partnered with local organisations and community leaders. Overall, contributors were involved between 90–180 min, per their preference and depending on the nature of their contribution.

In the second stage, we evaluated the PPI process. This involved contributors individually completing a brief survey (Appendix A). The survey was developed by the project team and had 11 items, focused on three areas: ease of contribution (4 items), accessibility (3 items), and safety/support (4 items). For ease of contribution, contributors were asked to rate whether they felt their contributions were listened to and valued on a Likert-type scale from ‘strongly agree’ to strongly disagree’. They were asked to report whether they felt there were different ways to contribute that worked for them, rated ‘yes’, ‘no’ or ‘partly’. They were asked to share any barriers or enabling factors associated with contributing. For accessibility, contributors were asked whether it was easy to share views in a way that suited them and whether they had the support they needed to participate fully, rated on a Likert-type scale from ‘strongly agree’ to strongly disagree’. They were asked to report whether there was anything that made participation difficult. Finally, for safe and supportive environment, contributors were asked whether they felt comfortable sharing their views without fear of judgment (i.e., relating to psychological safety) and whether the PPI approach was inclusive and respectful of different perspectives, rated on a Likert-type scale from ‘strongly agree’ to strongly disagree’. They were asked to share three words reflecting how the PPI felt to them and to describe what could have made them feel safer or more supported. Finally, using the Guidance for Reporting Involvement of Patients and the Public (GRIPP2-SF) checklist [11], the project team described and reflected on the PPI input and process.

All input was anonymised prior to analysis. Quantitative survey data were analysed descriptively (frequencies and percentages) using IBM SPSS Statistics, Version 28. Qualitative free-text responses from the survey were summarised and informed team reflections on the PPI process. Contributions from the PPI consultations were gathered into themes and categories in collaboration with our PPI-partner to ensure accuracy and align with our co-production approach. PPI contributions underwent a structured mapping process in which each response was coded to one of three predefined domains (ease of contribution, accessibility, and safety/support) or to an “other” category to capture emergent issues that did not fit the initial framework. The analytic process followed an iterative, inductive–deductive approach. First, two researchers independently reviewed all free-text responses to familiarise themselves with the data and to confirm or refine domain categorisation. Discrepancies in coding were discussed and resolved through consensus. The project team, including our PPI-partner, then synthesised the coded data to identify recurring patterns related to perceived enablers, barriers, and suggested improvements. Attention was given to similarities and differences between stakeholder groups (rehabilitation professionals and educators, individuals with lived experience, and technology innovators) to understand variation in perspectives. To enhance trustworthiness and ensure patient and public perspectives were accurately represented, illustrative quotations were selected by the PPI-partner. These short excerpts were used to ground analytic interpretations in contributors’ own words and to highlight the breadth of viewpoints within and across stakeholder groups.

PPI-contributors did not provide data as participants, and the thematic synthesis reflects consultation insights, not research data.

## 3. Results

In this section we describe PPI-contributor characteristics (Section 3.1) and present findings from the evaluation of the PPI process (3.2), documenting our use of the GRIPP2-SF checklist (Section 3.2, Appendix A).

### 3.1. PPI Contributor Characteristics

The 15 PPI-contributors (aged 28–72) included 8 women, 6 men, and 1 non-binary person. Ethnicities were White (9), Asian (3), Black (2), and Arab (1). There were six rehabilitation professionals (an occupational therapist, physiotherapist, orthotist, speech and language therapist, rehabilitation nurse, and medical registrar), five individuals with lived experience of serious injury, illness or disability requiring rehabilitation (including amputation, stroke, brain injury, hearing loss, and multiple sclerosis (MS)), two healthcare educators (with expertise in nursing, physiotherapy, and sports rehabilitation) and two technology innovators.

### 3.2. Evaluation of the PPI and Co-Design Process

In stage 1, evaluation survey findings from all 15 contributors (100% agreement on all items) reflected (i) the perceived ease of contribution (listened to, valued, and flexible approaches), (ii) high accessibility for sharing views (per participant preference, support provided), and (iii) the creation of a safe and supportive environment (non-judgmental, inclusive, and respectful). This is important in PPI since ease of contribution reduces cognitive and logistical barriers, allowing more people to participate meaningfully. Accessibility ensures that individuals with disabilities, language differences, or limited digital access are not excluded. Safety fosters trust and psychological comfort, especially for those sharing lived experiences. Support (e.g., compensation and emotional care) empowers contributors and sustains engagement. Areas where PPI could be better supported were reported by three contributors; all could be easily addressed in future PPI consultations. These included a preference for more than one meeting to discuss their views in more depth (n = 1), a contributor preference to respond by email, taking longer than a meeting due to the need to provide a written response (n = 1), and a contributor’s desire for their partner to join a future consultation meeting (n = 1).

The process functioned well as a mechanism for influence and inclusion. In addition to evaluating engagement methods, contributors highlighted the importance of understanding how their input was integrated into project decisions. Several noted that transparent communication about the influence of their contributions in the co-creation of a stakeholder engagement strategy to be used in future research would strengthen trust in the PPI process. The evaluation also suggested that sustainability of engagement, including adequate resourcing and opportunities for ongoing involvement, is critical to ensuring PPI remains meaningful rather than one-off. Finally, contributors valued opportunities for skill development, which enhanced their confidence to participate in future consultations.

In stage 2, The GRIPP2-SF checklist was applied and is included in the Appendix A. In this table, we clearly specify the PPI aim, methods, results, discussion/conclusions, reflections and critical perspective.

### 3.3. Identifying Overarching Challenges for Stakeholder Groups

PPI contributions were summarised in collaboration with our PPI-partner to ensure accuracy and confidentiality. The main challenges associated with training in rehabilitation technology are shown in Figure 1.

### 3.4. Detailing Considerations for Inclusivity and Accessibility

PPI-contributors from all stakeholder groups then identified important considerations for inclusivity and accessibility in patient-facing training relating to rehabilitation technologies. These were drawn into key areas of focus, shown in Figure 2 and described in Table 1.

### 3.5. Specific Challenges for Rehabilitation Patients

PPI contributors who had lived experience of serious injury, illness or disability requiring rehabilitation further detailed the key challenges for patients associated with training in rehabilitation technologies. These were grouped by the project team and PPI-partner into six categories: (i) Awareness & access, (ii) Overwhelm and timing issues, (iii) Complexity and usability, (iv) Cost and affordability, (v) Widening inequity, (vi) Practicality, and (vii) Confidence. These categories are described below and supported by illustrative quotes in Table 2.

(i)Awareness and Access

A significant proportion of people with lived experience of serious injury, illness or disability requiring rehabilitation remain unaware of the existence and potential benefits of advanced rehabilitation technologies such as robotic exoskeletons, virtual reality-based interventions, and neurofeedback systems. This lack of awareness is often compounded by limited access, particularly in rural or under-resourced settings where such technologies may not be available within local healthcare facilities or covered by insurance schemes. Consequently, many individuals are excluded from potentially transformative interventions due to informational and geographical barriers. In our consultations, contributors living with brain injury, stroke and MS identified that cognitive impairments (memory and attention) make it harder to seek out or retain information about available technologies. Our contributor with MS spoke of fluctuating symptoms, meaning patients may not always be in touch with services when new technologies are introduced. Our contributor with hearing loss suggested that information about technologies may not be communicated in accessible formats (e.g., captioning and sign language). A contributor who had experienced amputation felt that prosthetic and exoskeleton technologies are unevenly distributed, with advanced options often limited to specialist centres.

(ii)Overwhelm and Timing Considerations

The initial stages of recovery following neurological injury or physical trauma are frequently characterised by emotional distress and physical exhaustion. Introducing complex rehabilitation technologies during this vulnerable period can lead to cognitive and emotional overload, reducing receptivity and engagement. Appropriate timing is therefore critical: premature introduction may result in rejection or non-adherence, whereas delayed implementation may reduce the potential for optimal recovery. In our consultations, it was apparent from contributors who had experienced stroke, brain injury or amputation that early recovery often involves fatigue, confusion, and emotional distress, so introducing complex technology too soon can feel unmanageable. Our contributor living with MS indicated that relapses could make timing unpredictable; technology introduced during a flare-up may therefore feel unusable.

(iii)Complexity and Usability

Many rehabilitation technologies necessitate the acquisition of new skills, interaction with sophisticated interfaces, and adherence to specific operational protocols. For people experiencing cognitive deficits, motor impairments, or fatigue, such demands can present substantial challenges. Poor usability not only increases frustration and non-compliance but also diminishes therapeutic efficacy. Designing systems with user-centred principles is thus essential to facilitate sustained engagement and maximise clinical outcomes. For our contributor living with hearing loss, audio-based instructions or feedback systems were described as less effective, requiring alternative design. In our consultations, individuals living with stroke, brain injury and MS highlighted that interfaces requiring multitasking or fine motor control are especially difficult for those with hemiparesis or cognitive deficits. Our contributor with hearing loss described audio-based instructions or feedback systems being less effective, requiring alternative design.

(iv) Cost and Affordability

The financial burden associated with advanced rehabilitation technologies remains a major barrier to adoption. High device costs, inconsistent or absent insurance reimbursement, and additional expenses for training or travel to specialised facilities place considerable strain on patients and families. These economic constraints are particularly consequential for individuals already experiencing income disruption because of disability or prolonged recovery. In our consultations, a contributor with MS discussed the long-term, progressive nature of MS, which means ongoing costs accumulate, straining finances. An individual who had experienced amputation shared concerns that prosthetics and robotic aids are notoriously expensive, often not fully covered by insurance.

(v) Widening Inequity

The uneven distribution of rehabilitation technologies across geographic and socioeconomic strata can exacerbate existing health disparities. People living in affluent regions or private healthcare systems may be more likely to access cutting-edge interventions, while those in underfunded or public systems often rely on conventional or outdated methods. Such inequities not only influence recovery trajectories but also perpetuate broader systemic imbalances in healthcare provision and outcomes. In our consultations, a contributor living with hearing loss raised concerns that those without access to interpreters or inclusive services are disproportionately excluded.

(vi)Practicality and Integration into Daily Life

Even when accessible and affordable, rehabilitation technologies may pose practical challenges that limit sustained use. Devices may be bulky, require frequent calibration, or necessitate attendance at specialised centres. For people managing competing responsibilities, such as employment, caregiving, or limited mobility, these logistical demands can outweigh perceived therapeutic benefits. Successful integration requires consideration of real-world contexts and patient lifestyles. In our consultations, a contributor living with MS reminded us that devices that require sustained energy or travel are challenging due to fatigue and mobility limitations. An individual who had experienced amputation reported that devices may be heavy, require calibration, or not fit seamlessly into daily routines.

(vii)Confidence and Self-Efficacy

For people with lived experience of serious injury, illness or disability requiring rehabilitation, confidence in their ability to effectively use rehabilitation technologies is a critical determinant of engagement. Individuals with limited technological literacy may experience apprehension, fear of failure, or embarrassment, which can undermine motivation and adherence. Structured support, peer modelling, and progressive skill development are essential to enhance self-efficacy and promote long-term participation in technology-assisted rehabilitation. In our consultations, contributors with stroke and brain injury shared that fear of failure is heightened when cognitive or speech difficulties make communication with clinicians harder. Similarly, for a contributor living with hearing loss, communication barriers reduced their confidence, especially if training relied on verbal guidance. An individual who has experienced amputation discussed that learning to use prosthetics requires persistence and that frustration with fit or function can undermine motivation and confidence.

### 3.6. Specific Needs of Rehabilitation Patients

With knowledge of other stakeholder views, PPI contributors with lived experience of serious injury, illness or disability requiring rehabilitation identified and described the factors that they deemed to represent their key needs related to receiving and engaging with training in rehabilitation technologies (see Table 3).

## 4. Discussion

We engaged with 15 rehabilitation professionals and health educators, people with lived experience of serious injury, illness or disability requiring rehabilitation, and technology innovators to determine their views on factors related to the inclusivity and accessibility of training on rehabilitation technologies to individuals with sensory, cognitive and physical impairments and (b) patient needs associated with training in rehabilitation technologies. Perceptions of contributors towards our PPI approach highlighted that the approach maximised ease of contribution, accessibility, safety and support. This is essential to ensure meaningful engagement and equitable participation, reduce power imbalances, and enhance the quality and impact research [15]. If technology is to be equitable, usable and sustained, then inclusive, user-centred design is critical, and this has been highlighted elsewhere [16,17].

Our PPI work makes a distinctive contribution by advancing a multidimensional framework for inclusivity and accessibility in patient-facing training for rehabilitation technologies. Drawing directly on PPI consultations, we move beyond conventional accessibility discussions that focus narrowly on physical or sensory adjustments to encompass cognitive accessibility, motivation and engagement, instructional design, staff competence, and policy alignment. This holistic approach highlights the paradox that rehabilitation technologies, while designed to support recovery, may themselves create barriers if training is not inclusively designed. By embedding principles of Universal Design for Learning [12] within rehabilitation contexts, foregrounding the importance of staff empathy and competence, and situating accessibility within broader legislative frameworks such as the UN CRPD [14], our work positions inclusivity as both a practical and systemic imperative. In doing so, it reframes accessibility not simply as a technical requirement but as a patient-centred, empowering, and policy-embedded practice that ensures rehabilitation technologies are usable, meaningful, and equitable for diverse populations.

The PPI process facilitated the collaborative design of the stakeholder engagement strategy (Appendix A) by bringing together individuals with lived experience of serious injury, illness or disability requiring rehabilitation, healthcare professionals and educators, and technology innovators. This achieved its purpose in identifying the key challenges and barriers to engaging with training on rehabilitation technologies and the enabling factors that must be considered in training design and delivery. Our findings support prior research in this field, which emphasises the importance of ‘co-design’ and direct stakeholder engagement in the development of rehabilitation technologies [2]; here, we advocate for stakeholder engagement in education and training relating to such technologies. Through iterative consultation and refinement, these diverse perspectives were integrated into a shared framework, developed by the project lead (HB), in collaboration with our PPI-partner (VAF). Content was reviewed by seven collaborators (including co-authors AA and MH, and five other individuals from a wider research team). The team brought lived experience, as well as expertise in health and social care services, behavioural science, rehabilitation (physical and mental health), innovation and technologies. A stakeholder engagement strategy (Appendix A) was subsequently structured using the Implementation STakeholder Engagement Model (I-STEM), ensuring that stakeholder roles, engagement strategies, and implementation milestones were systematically aligned with theory-informed guidance from the I-STEM Toolkit [18]. The development process was therefore grounded in implementation science, which provides a structured, theory-informed approach to stakeholder engagement during the design and implementation of health and social care innovations [18]. The strategy has informed our next steps, including an evidence review, a survey study, and a series of stakeholder consultation events, which will provide deeper insights into the education and training needs of rehabilitation professionals, patients and technology innovators associated with rehabilitation technologies. Based on the PPI contributions here, our PPI partner created a public contributor’s guide to ‘What Patients Want’, which could be used by rehabilitation professionals and technology innovators to ensure that rehabilitation technologies (and the way in which they are implemented) are inclusive and accessible to individuals with sensory, cognitive and physical impairments.

In summary, this work has influenced the next stage of our study in three ways. First, it has confirmed the appropriateness of our PPI approaches and supported us in embedding equitable engagement principles within our work. Feedback on the PPI process itself-emphasising accessibility, psychological safety, and reduced power imbalances—will be incorporated as standards for ongoing stakeholder engagement. These principles will underpin iterative consultation, promote meaningful participation, and enhance the translational impact of the study as it progresses into implementation-focused phases. We frame inclusivity as a systemic and rights-based imperative rather than just a technical or design concern; this aligns with the stance taken by other researchers who argue for the importance of embedding human rights approaches in PPI [19]. Second, the PPI findings directly shaped the co-creation of a theory-informed stakeholder engagement strategy, which will guide forthcoming evidence review, survey work, and stakeholder consultation events. This ensures that subsequent research activities reflect the priorities, language, and accessibility needs identified by people with lived experience, rehabilitation professionals, and technology innovators. Third, this work was the first step in establishing priorities for education and training relating to rehabilitation technologies. We gathered insights into contributors’ key challenges to engaging with training in rehabilitation technologies, which align with findings from prior research. These include awareness and access, cost and affordability [20], issues with complexity, overwhelm and timing [2], widening inequities [21] and confidence or self-efficacy levels [2]. Our contributors highlighted what patients want from training in rehabilitation technologies, enabling the creation of a ‘public contributor’s guide’ focused on inclusivity, accessibility, safety, and ease of participation. This guide will inform the design of training materials and data collection processes to better accommodate sensory, cognitive, and physical impairments.

### Limitations

Although this PPI process generated valuable insights, several limitations should be acknowledged. First, the sample size was small (n = 15) and comprised a convenience group of rehabilitation professionals, individuals with lived experience, and technology innovators. This was an appropriate sample size to meet our PPI aims, and other co-design studies have engaged a comparable or smaller number of public contributors [22,23,24]. Nonetheless, the perspectives captured may not represent the full diversity of rehabilitation stakeholders, particularly those from under-represented groups, non-English speakers, or people with profound cognitive or communication impairments.

Second, although contributors brought varied expertise, their experiences were largely drawn from high-income health and social care contexts, which is a known limitation of published PPI activity [25]. Although our work does focus on experiences in the UK, it may limit the transferability of findings to settings with different resource constraints, digital infrastructure, or cultural expectations around rehabilitation technologies and training.

Third, the qualitative analysis was based on brief free-text survey responses and consultation input collected individually in diverse ways (email, telephone, and video call) rather than in-depth interviews or focus groups. While this approach supported accessibility and ease of participation, it constrained the depth of insight and limited opportunities to probe emerging issues. Additionally, mapping responses to predetermined domains may have introduced a degree of researcher framing, potentially overlooking nuanced or cross-cutting experiences.

Finally, while a PPI partner supported synthesis and selection of illustrative quotations, the involvement of broader contributors in data interpretation was limited to feedback stages rather than shared analytic decision-making throughout. This may have reduced opportunities for co-analysis and could reflect a residual power imbalance in knowledge production. Future PPI cycles involving larger, more diverse samples, multimodal data collection (e.g., interviews or stakeholder focus groups and workshops), and co-analysis strategies could help address these limitations and further strengthen the inclusivity and validity of stakeholder-informed research on rehabilitation technologies.

## 5. Conclusions

Advancing inclusive and accessible training in rehabilitation technologies necessitates a holistic framework that integrates physical, sensory, and cognitive accommodations with inclusive pedagogical practices and user-centred technological design. The application of universal design principles, strategic deployment of assistive technologies, and cultivation of competence and confidence in technology use are critical to establishing equitable training environments. Such inclusivity not only enhances access and engagement but also strengthens the effectiveness of rehabilitation technologies, thereby promoting independence and safeguarding the dignity and rights of individuals with disabilities. To extend this work, we propose a scoping review of the published literature to systematically examine the education and training requirements of healthcare professionals in relation to rehabilitation technologies. Complementing this, a survey of healthcare professionals will be conducted to identify training priorities, while qualitative interviews and focus groups with diverse stakeholders—including individuals with lived experience, healthcare providers, and technology innovators—will provide in-depth perspectives. Collectively, these findings will inform the design, development, and implementation of future training initiatives in rehabilitation technologies.

## Figures and Tables

**Figure 1 ijerph-23-00013-f001:**
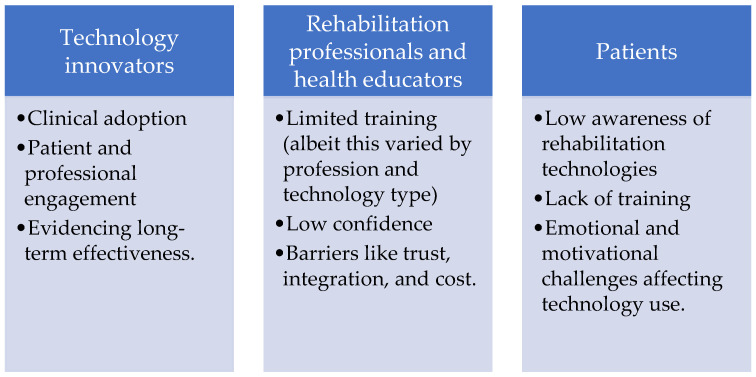
Overarching challenges for each stakeholder group.

**Figure 2 ijerph-23-00013-f002:**
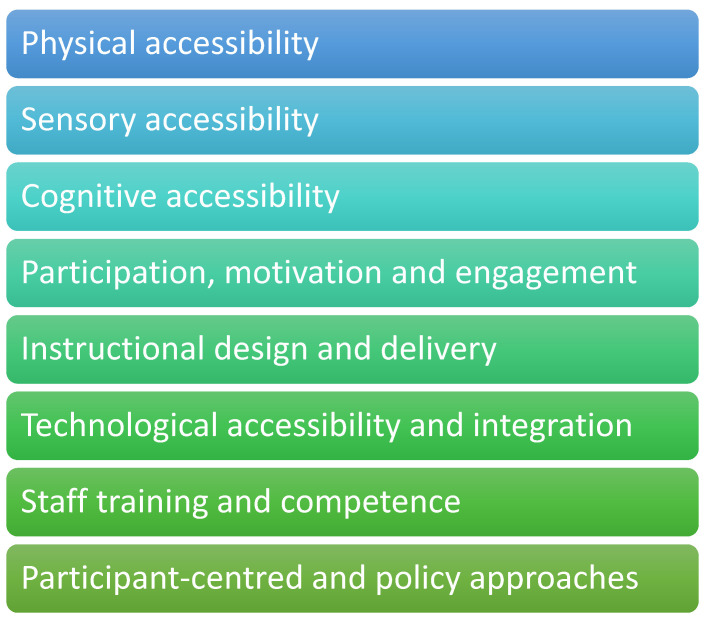
Factors supporting inclusive and accessible patient-facing training.

**Table 1 ijerph-23-00013-t001:** Description of factors supporting inclusive and accessible patient-facing training.

Consideration	Description
Physical Accessibility	Inclusive training environments must be physically barrier-free, with ramps, wide doorways, accessible restrooms, and adjustable workstations. Rehabilitation technologies should be positioned for safe, independent use by wheelchair users or those with prosthetics, with equipment adaptable to varied body sizes and functional levels. Emergency procedures must also account for mobility limitations. “When I arrived, the doorways were too narrow for my prosthetic crutches.” (amputation)
Sensory Accessibility	Training must accommodate visual and hearing impairments. Materials should be available in Braille, large print, audio, and screen-reader-compatible formats, supported by high-contrast displays, tactile markers, and verbal descriptions. For hearing impairments, captioning, sign language interpretation, written instructions, and subtitled tutorials are essential, alongside visual alerts and diagrams. “The trainer kept explaining things verbally, but without captions or an interpreter I missed half of it. I needed visual instructions to feel included.” (hearing loss)
Cognitive Accessibility	Clear communication and structured learning are vital for those with cognitive impairments. Instructors should use plain language, multimodal delivery (verbal, visual, and hands-on), and break complex procedures into sequential steps with pictorial guides. Extra time for repetition and distraction-free environments further support comprehension and focus. “After my injury, I struggle to follow long explanations. Breaking the steps into simple pictures and repeating them slowly made all the difference.” (brain injury)
Participation, Motivation, and Engagement	Motivation is strengthened when training is meaningful, empowering, and aligned with personal rehabilitation goals. Interactive approaches—such as gamification, simulations, and real-world tasks-enhance engagement, while peer support fosters belonging. For those with cognitive or sensory impairments, clarity, positive feedback, and emotional support are critical. Staff empathy and adaptability build trust, while sensitivity to timing and mood prevents disengagement. Co-designing training with participants ensures relevance and ownership. “I stayed motivated when the exercises connected to real tasks, like making a cup of tea. It reminded me why I was doing this in the first place.” (stroke)
Instructional Design and Delivery	Grounded in the Universal Design for Learning (UDL) framework [12], inclusive training should provide multiple means of engagement, representation, and expression. Learners benefit from individual pacing, continuous accessible feedback, and opportunities for peer collaboration, which reinforce confidence and inclusion. “Because my fatigue varies day to day, having the option to go at my own pace meant I could keep learning without feeling defeated.” (multiple sclerosis)
Technological Accessibility and Integration	Rehabilitation technologies must themselves be accessible, complying with WCAG 2.1 [13] and supporting assistive tools such as screen readers, voice control, and adaptive switches. Customisable outputs and integration of devices like speech-to-text systems ensure full participation, positioning technology as an enabler rather than a barrier. “The software had no subtitles, so I couldn’t follow the video prompts. Technology should help me, not shut me out.” (hearing loss)
Staff Training and Competence	Instructor competence is central to inclusion. Professionals should be trained in disability awareness, adaptive teaching, and assistive technology use. Empathy, flexibility, and collaboration foster supportive environments, while ongoing professional development ensures responsiveness to emerging inclusive practices. “She [therapist] took time to listen and adapt the instructions, that gave me confidence. Without that I would have given up.” (stroke)
Participant-Centred and Policy Approaches	Training should be guided by participant-centred principles, tailoring goals and methods to individual needs. Institutional and policy frameworks must align with disability legislation, such as the United Nations Convention on the Rights of Persons with Disabilities (CRPD, [14]), embedding inclusion as both compliance and culture, thereby promoting empowerment and equal participation. “When I was asked about my personal goals, I felt respected. It wasn’t just about ticking boxes then getting me out of there, it was about my life.”

**Table 2 ijerph-23-00013-t002:** Patients’ challenges and training needs with illustrative quotes.

Key Challenges	Illustrative Quotes
Awareness and access	“As a patient, one of the biggest challenges is even knowing what’s available, where it’s available from, what use is it to me and the cost. Most of us are never told about rehabilitation technologies. A lot of the time we generally only find out by chance, or we research ourselves, or through other people. This creates a real inequality, because the people who would benefit most may never get the chance, or may never know what’s out there for them.” (amputation) “They’re too busy doing this and that, nothing said and then you hear about something but where would I find it?” (multiple sclerosis)
Overwhelm and timing issues	“When you’re first adjusting to life-changing disability or injury, you’re often overwhelmed—physically, emotionally and mentally. You generally deal with what’s happened that day, what’s the most important and what capacity you have as well. That’s not the time to be handed complex technologies without support which can and does happen. If people are struggling with low mood or pain or even just after a life changing injury or complex medical procedures, the technology can feel like another burden rather than something empowering. Being simply to press that button and it does this and that does something else when you’ve just had surgery a couple of days ago, you’re not going to be able to take it all in and understand how it can help or what good it’s going to do.” (amputation) “It’s changed your life, rushing around, seeing this person, that person, surgery, hospitals. All I want to do is sit and cry about what I have lost. Not the time to start saying ‘want to try this new thing?” (stroke)
Complexity and usability	“A lot of the technology looks amazing, but it often feels designed for clinicians or engineers, not patients. If something is too complicated, has too many steps, or isn’t intuitive, it becomes frustrating very quickly. Many people now have ‘smart watches’ that can do great things but for many they’re just too complicated to use on a daily basis or to simply just to use them or don’t understand how to interpret the results from the tech to send on to the healthcare professional. People will just give up, which wastes the potential of the technology. It is also a waste of money if you don’t get good use out of it or if you’re not using it to its full potential. Not everyone is good at using tech whatever age you’re at which can be a big barrier to many.” (amputation) “Do they make what people want? Have they asked what we need? It’s so complicated, even those apps with all the different functions and you don’t use it, but if they asked us, we’d have said get rid of this and that. I don’t want it. Just give me the basics.” (brain injury)
Cost and affordability	“Many pieces of technology are expensive and not funded, which means only some people can access them. That leaves a lot of people feeling left behind or without a tool that could be extremely valuable which could improve someone’s quality of life, which doesn’t seem fair.” (amputation) “If you don’t have the money, are they going to give it to you? Or no, more likely you’ll be left on the side?” (hearing loss)
Widening inequity	“A lot of tech can and does rely on the internet, WiFi or integration with smart phones or tablets. Not everyone has access to things like smart phones or even the internet … which makes it unfair and unjust for those who don’t have access to the internet or IT.” (amputation) “The ones that make it, they’ve got all that fancy stuff, but I like it simple. I don’t know what connects to what, and they use words I don’t know about. Is someone going to set it up, fix it when it goes wrong?” (hearing loss)
Practicality	“Some of the tech doesn’t fit well into real life. They might work brilliantly in a lab, but in a patient’s home or community setting, they can be too bulky, need too much space, or don’t integrate easily with daily routines.” (amputation) “I work from home and need to be able to use it without it becoming a problem due to lack of space.” (multiple sclerosis)
Confidence	“For some people, especially if they’ve had bad experiences before, there can be a lack of confidence. You worry—will it work for me, or will I fail at it? That sense of not wanting to ‘get it wrong’ can be a barrier for many people. Also, confidence in using the tech itself. If you’re not a confident tech person then adding in a wearable or another piece of tech that you just don’t get is a waste of time, money and effort, especially if you’re shown how to use it once and then you’re left to use it yourself.” (amputation) “I don’t know if I could. You know it’s so much, so much going on and if I won’t get it straight away then are they gonna sit down and go over and over? You already think you can’t do anything, and they probably think you can’t do anything.” (stroke)

**Table 3 ijerph-23-00013-t003:** Public contributor’s guide to ‘What Patients Want’.

Item	Detail and Requirements
Plain language & step-by-step learning	People need explanations in clear, non-technical, easy-to-understand language-without the use of any form of jargon or medical/technology specific wording. Training should be broken down into simple, manageable steps rather than a huge instruction manual that overwhelms you from the start or steps that miss important points.
Repetition and reinforcement	When you’re adjusting to disability, fatigue and brain fog can make it hard to retain new information. So having either regular refreshers or repetition of the training are essential, not a one-time explanation.
Hands-on practice	Seeing and trying the technology with guided support is key. People learn best when they can practice at their own pace, with someone patient enough to repeat things as many times as needed.
Tailored to individual needs and using different forms of learning	Training and education should recognise that everyone’s starting point is different in terms of ability, confidence, mood and energy. Some may need very short sessions, while others may want more in-depth training. Using different forms of training is important; some people learn by watching a video and following along, others prefer watching someone physically use something, and others prefer step by step written guides or pictorial step guides.
Individualised pace	Training needs to be adapted to each person’s abilities, confidence level, and learning style. For example, someone with limited hand movement will need training adapted for that. Or if you learn slower and need more repetition, you don’t want to be in a group that speeds through everything, and you’re left behind.
Psychological readiness	Training needs to consider mental health. If a person is anxious, depressed, or still adjusting to their condition, that will affect how much they can take on. Offering emotional support alongside technology training can often make a big difference.
Ongoing support, not one-off	It’s not enough to show someone once and leave them to it. People need follow-up, ongoing support and a way to ask questions when problems come up at home. You can’t expect someone to hear a trainer say what you need to do on one occasion and then use the tech yourself. That’s when people give up and stop using things.
Peer learning	Sometimes the best training comes from other people who’ve already used the tech. Seeing someone like you use the tech successfully can very often build confidence and make it feel more achievable for you to use it.
Family and carer involvement	Often carers or family members are the ones helping patients use technology at home. Training should include them too, so everyone feels confident and supported. It can also help if someone is with you when you’re learning a new piece of tech because they will most likely remember instructions as they’re generally not in any pain or fatigue.
Accessibility of information	Training resources should be available in different formats—written guides, videos, easy-read, and online resources—so people can choose what works best for them. “We’re not all clones, we don’t all learn the same way!”.
Cultural & social awareness	Training should also consider cultural factors and health literacy. Some people may feel less confident speaking up if they don’t understand or come from a culture where it’s impolite to ask or disturb a class or if they are used to their husband/partner speaking up for them. Training should always be inclusive and sensitive.
Visual and practical learning	Not everyone learns well from a written manual. Training should include demonstrations, short videos, diagrams, podcasts and hands-on practice. Real-life examples of how someone might use the technology at home or at work make it easier to understand.
Supportive environment	The way training is delivered matters as much as the content to people. We just need an encouraging, non-judgmental environment where we don’t feel embarrassed if we ‘get it wrong’ or we need something explained again.
Clear information on benefits and limits	We need realistic expectations! Training should explain what the technology can do, but also what it can’t. That way, people aren’t left disappointed or feeling like they’ve failed if it doesn’t ‘fix’ everything.

## Data Availability

All data are provided within the manuscript.

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
