# Peer review of "Co-Designing an Inclusive Stakeholder Engagement Strategy for Rehabilitation Technology Training Using the I-STEM Model"

_ijerph, 2025, doi:10.3390/ijerph23010013_

Round 1

Reviewer 1 Report

Comments and Suggestions for Authors
  • Overall, the manuscript presents an interesting broad ranging survey with different stakeholder groups associated with developing, prescribing, and using rehabilitation technologies. Several sections within the manuscript could be improved for clarity and conciseness.
  • A major concern is that the paper has limited details about how the questions in the survey used for this study were customized for each stakeholder groups or how were they phrased so that they would be appropriately interpreted by these groups. Technology developers have very different needs from the design process than the actual end users of the technologies (which could be individuals with disabilities or clinicians depending on the type of technology). It was unclear if combining the stakeholder groups together did not adequately bring out issues and needs specific to individual stakeholder groups. In other words, the survey may have missed capturing important details from specific stakeholders.
  • It is recommended to include the survey questions as an appendix for the readers
  • Some of the results from this research seem to be what a textbook on accessibility and inclusivity may describe. It is good that the researchers have found similar factors from their interviews but the novelty or contribution to literature is harder to ascertain from the results as presented in the manuscript.
  • From the description of the survey it appears to be generally focused on gathering needs from use of rehabilitation technologies. This is important aspect to evaluate but it differs from the stated aims of the study to evaluate needs specific to “training”. It was unclear how the survey questions specifically asked about training needs rather than need for use of the technologies.
  • In the section on author affiliations re-check the formatting guidelines of the journal about including contact details of all authors or only the corresponding author.
  • There are limited details about specific methods used with the survey data from the different stakeholder groups to generate the themes as described.
  • The purpose of Table 1 is unclear. Unless it adds or differentiates from anything beyond the similar sections of the paper, I suggest not to include it and integrate the contents in different places of the paper.
  • The section on challenges for stakeholder groups is very important but the manuscript does not adequately describe these challenges. These challenges would be unique and distinct for the specific sample of stakeholders in this study. It would be helpful to read quotations specific to those challenges and additional details about them.
  • In Figure 2, the blocks represent useful classification of factors. However, the presentation in a circular format and the arrows do not make sense as described in the paper. It was unclear how the individual blocks feed into the subsequent ones.
  • The section 3.4 can be significantly reduced in length since the factors described are textbook domains of concerns when addressing accessibility and inclusivity. Specific quotations can be helpful for any novel or important (as identified by the stakeholders) factors. It would also be interested to read which factors are relevant to the individual stakeholder groups. The section reads more as a set of guidelines related to rehabilitation techniques rather than how the responses from the stakeholders were used to generate them. Making this section concise will improve readability of the paper and will highlight other important results from the research.
  • The challenges and needs of “rehabilitation patients” are discussed in detail and can be a good contribution of this paper. The implications of these findings could be discussed in the discussion section in detail, perhaps finding parallel with other published resources.
  • Conclusion sections typically are short and concise. Separate out points of discussion out of the conclusion section into their own discussion section. Also, in this discussion section, have a broad ranging discussion of the major results from the paper and discuss any limitations in the study design or sample.
  • The manuscript does not adequately identify the co-creation process used for developing the stakeholder engagement plan. Since this plan will be described somewhere else, this paper can make it sound as a future work or ongoing work. The authors may also describe which sets of data derived from this study will be synthesized for generating this plan and the specific need for such a plan within the context of the hospital system. To emphasize the need for the study presented here the authors may mention that generating the stakeholder engagement plan is an ultimate goal of this research.
Comments on the Quality of English Language
  • Line 34: Instead of “ease healthcare demands” use phrases such as "ease the burden of healthcare demands."
  • Line 65, consider using a different phrase instead of “sharing power and responsibility” since it seems hierarchical. Possibly use “agency”
  • Lines 119-123: The preferences seem to be towards the methods employed in this study and not necessarily towards the PPI process
  • The manuscript has few places where person-first or identify first language is not used while referring to individuals with disabilities. It is recommended to review this language. 

Reviewer 2 Report

Comments and Suggestions for Authors

Thank you for your submission. This manuscript offers a thoughtful integration of both patient and clinician stakeholders to inform, identify, and move forward priorities in tech design.

Abstract: 1-20, to open the sentence “… contributors identified …” consider adding a brief statement on the methodology used for theme development. This abstract is not broken down in the traditional sense of intro, methods, results, conclusion, so a brief statement on the methods is still warranted for a scholarly manuscript abstract.

Methods: 2-23 The statements at the conclusion make it clear that this manuscript does not reflect research, so you may want to clarify here that PPI contributors did not provide data as participants, and that the thematic synthesis reflects consultation insights, not research data, as quotes are included and conclusions drawn.

For repeatability, how long did the contributor participate (such as time of meeting, or a range)?

3-99, analysis section is missing on the methods used to analyze, as a manuscript, it requires more clarity on the qualitative design and analysis as to how data were “gathered into themes and categories.” Please describe what specific qualitative method was used (e.g., thematic analysis, content analysis). Please include how disagreements were rectified.

Did the PPI partner co-code or simply review synthesized categories?

Results: Page 8 Section 3.5 (Challenges and quotes): Excellent content, but lengthy and does not provide much insight and meaning without the reader understanding the background of the age, condition, and technology use. Is it possible, at a minimum, to include the patients’ medical condition to provide a richer picture of their lived experience? 

Discussion/Conclusion: 11-288 ”The PPI led to the co-creation of a stakeholder engagement plan…”  This manuscript is missing a discussion section but still requires implications within the conclusion. I personally am a developer, but I’m not quite sure how to use this information of “structured, theory-informed approach” in design. To the readers and for the impact of the manuscript to be published, what’s are the concrete take aways?  Please strengthen the translational impact by providing 2–3 concrete examples of how insights will shape the next phase.

Please also add a limitation section to complete the manuscript.

Reviewer 3 Report

Comments and Suggestions for Authors

I enjoyed reading the manuscript. I think it is an important contribution to the field. I agree that adoption of Rehabilitation Technology remains uneven due to challenges like limited stakeholder training, lack of professional confidence, and low integration into practice. Patient and public involvement (PPI) is the essential first step.  I only have a couple of suggestions:

  1. Regarding Table 1 (p. 3&4), I am not sure this whole table is needed since it summarizes some formation already shared (aims, methods). Perhaps it could only focus on the results, discussion and reflections sections.
  2. Regarding Figure 3 (p. 9), it looks like a table in the current form, not a figure.
  3.  Regarding point # 3 "Hands-on demonstration & repetition" on p. 9, I suggest using the word "practice" to avoid using repetition since it was mentioned in the previous point.
